# Changes in Fatty Acids Profile, Health Indices, and Physical Characteristics of Organic Eggs from Laying Hens at the Beginning of the First and Second Laying Cycles

**DOI:** 10.3390/ani12010125

**Published:** 2022-01-05

**Authors:** Lukáš Zita, Monika Okrouhlá, Ondřej Krunt, Adam Kraus, Luděk Stádník, Jaroslav Čítek, Roman Stupka

**Affiliations:** Department of Animal Science, Faculty of Agrobiology, Food and Natural Resources, Czech University of Life Sciences Prague, 165 00 Prague, Czech Republic; okrouhla@af.czu.cz (M.O.); krunt@af.czu.cz (O.K.); krausa@af.czu.cz (A.K.); stadnik@af.czu.cz (L.S.); citek@af.czu.cz (J.Č.); stupka@af.czu.cz (R.S.)

**Keywords:** age, atherogenic, fatty acid, laying hens, organic, thrombogenic

## Abstract

**Simple Summary:**

Eggs are some of the most valuable components of the human diet. Nowadays, the “trend” is for humans to follow healthy lifestyle behaviours. Moreover, special emphasis is placed on animal welfare and in the handling of animals. Thus, hen-laying cycles are more often extended (or ended) by moulting, followed by another cycle, instead of depopulation. Egg quality during the life of a hen was monitored to optimise the hen’s age of the cycle ending (in terms of fatty acid composition and its impact on human health). The present study compared the egg qualities from hens, of two laying cycles, regarding the fatty acids profile, and hypocholesterolemic, atherogenic, and thrombogenic indices. Younger hens had higher albumen and yolk indices, Haugh units, and eggshell strength; monounsaturated fatty acids (MUFAs) were lower and saturated fatty acids (SFAs) were higher. The polyunsaturated fatty acid (PUFA) n-6/n-3 ratio, saturation, atherogenic, and thrombogenic indices were significantly lower in the egg yolks from older hens compared to younger layers. These findings prove that it is practical to utilize them in the organic farming system during a period of two years.

**Abstract:**

The present study compared the fatty acid profile and some physical parameters of eggs from hens reared according to the organic system at the beginning of the first and second laying cycle. A total of 1080 eggs were analysed at the beginning of the first (from the 28th to 30th week of age) and the second (from the 78th to 80th week of age) laying cycle. It was found that the hen ages influenced the egg weight, egg surface area, yolk proportion, and eggshell colour. Albumen and eggshell proportion, albumen, yolk index, Haugh unit score, and eggshell strength were lower in eggs from older hens compared with those produced from younger layers. Monounsaturated fatty acids were found in higher amounts than saturated fatty acids and polyunsaturated fatty acids in egg yolks of eggs from layers only at the beginning of the second laying cycle. The PUFAn-6/n-3 ratio, saturation, atherogenic, and thrombogenic indices were significantly lower in the egg yolks from older hens compared to younger layers. These findings (regarding the eggs from the older ones) prove that it is practical to utilize them in the organic farming system during a period of two years.

## 1. Introduction

The welfare of laying hens is an important topic, not only among scientists, but also among non-expert communities. The type of housing is a main concern, where the trend has been to follow alternative or different methods than cage housing [1]. Emphasis is also placed on the quality of production, specifically on the quality of eggs, which are an integral part of the human diet. The quality of eggs can be influenced in many ways, such as by genotype [2], feeding [3], hen’s age [4], or housing system [5]. Regarding the age of hens, there is a tendency to extend the laying cycle to reduce costs (depopulation) and improve benefits (pullet purchasing) of the breeding [6]. Considering the housing of hens in the EU, there were 55.6% of hens in enriched cages, 25.7% housed on deep litter, 14.1% in free-range systems, and 4.6% were reared in organic systems [7]. While comparing these systems in terms of egg quality, the differences in each component of eggs are scientifically described in many studies [3,8]. The internal quality of eggs depend on the composition and count of nutrients—high-quality proteins, carbohydrates, minerals, vitamins, and lipids, such as phospholipids and polyunsaturated fatty acids (PUFAs) for the human diet [9]. Fatty acids are elements of cell membranes and cell organelles, where they influence the permeability of nutrients to the cells of the human organism [10]. The content of fatty acids is about 26.6 g/100 g of yolk. Monounsaturated fatty acids (MUFAs)—46.9% and polyunsaturated fatty acids—22.4% are the dominant ones, whereas saturated fatty acids (SFAs) constitute the remaining 30.7% [11].

The human organism is not capable of synthesizing long-chain n-3 PUFAs. Therefore, these fatty acids must be provided by the diet [12]. In general, eggs are among the best sources of n-3 PUFAs, which are essential, especially for growing individuals [13]. Eggs naturally do not contain any eicosapentaenoic (EPA) and docosahexaenoic (DHA) fatty acids; moreover, they are a poor source of linoleic acid [14]. However, the amount and profile of fatty acids in egg content can be easily affected by feed, e.g. by grass addition [15]. From a health point of view, an inadequate intake of n-3 PUFAs, specifically of DHA, harm the brain growth and functional parameters of infants [14]. Consumption of foods that are rich in unsaturated fatty acids (especially in PUFAs n-3) is associated with a reduction of cardiovascular disease risk [16]. Other health benefits of n-3 fatty acids include their neuroprotective properties, which positively affect cognitive functioning, connected with aging [17]. Considering the nutritional value, the ratio between n-3 and n-6 fatty acids range from 1:1 to 1:4 [18].

Nevertheless, to our knowledge, a limited number of scientific literature focuses on the effect of age (or laying cycle) on the fatty acid profile and human health indices of eggs that come from organic farming. Authors, such as Lopez-Bote et al. [15], often focus on free-range housing, with the possibility of pasture, but do not focus on organic farming. We hypothesize that eggs from hens in the second laying cycle will show a lower internal and higher external quality than eggs from the first laying cycle. We further hypothesize that eggs from the second laying cycle will have a different profile of fatty acids in comparison with eggs from the first laying cycle.

The objective of this study was to investigate the differences between fatty acid compositions, health indices, and some external, internal quality characteristics of organic eggs from laying hens produced at the beginning of the first and second laying cycles.

## 2. Materials and Methods

### 2.1. Animals and Experimental Design

The genotype Lohmann Brown was used in this study. Breeding conditions met the requirements set by Council Regulation (EC) no. 834/2007 of 28 June 2007 (Council Regulation (EU) no. 2018/848 of 30 May, 2018, on organic production and labelling of organic products, and repealing Council Regulation (EC) no. 834/2007) on organic production and labelling of organic products, and repealing Regulation (EEC) no. 2092/91. All pullets were obtained from a certified organic poultry breeder (Austria) at 18 weeks of age (October). All birds were raised according to certified organic production methods immediately after hatching. The hens were provided with outdoor free-range throughout the whole year. Feeding by a feed mixture for hens was applied from the hen age of 20 weeks.

Feed mixture was certified organic; its constituents were wheat, organic corn, organic peas, organic soybean cake, organic sunflower cake, calcium carbonate, organic rapeseed cake, corn gluten, organic wheat, organic soybean oil, potato protein, and mono-calcium phosphate. Feed mixture composition was specially designed for this study by experts for hen nutrition, from feed company Sehnoutek a synové Ltd. (Voleč, Czech Republic), to fully cover the requirements of hens throughout the whole study. Chemical composition of diet and its major fatty acids profile are shown in Table 1.

The hens had unlimited access to feed and water, so feeding and drinking were ad libitum. No drugs or medicaments, such as antibiotics or production stimulants, were used. At 18 weeks of age, the lighting regime was set to 10 h of light. The period of light regularly extended to 16 h at the age of 26 weeks, and it was maintained until the end of the study. Artificial light was used only when the natural light was shorter than 16 h. The environmental conditions were set to meet the regular criteria for poultry farming.

### 2.2. Egg Quality and Fatty Acid Analyses

Egg quality analyses were performed at the Department of Animal Science of the Faculty of Agrobiology, Food, and Natural Resources of the Czech University of Life Sciences Prague (Czech Republic). A total of 1080 eggs (i.e., of the layers) were analysed in the trial. The eggs were collected three times at the beginning of each laying cycle (180 eggs at 7-day intervals from the 28th to 30th week of age and from the 78th to 80th week of age). The eggs were collected fresh and were immediately subjected to various physical analyses. Parameters taken on each egg were their weight, egg shape index, the yolk, albumen, eggshell percentage to egg, Haugh units, yolk, albumen, shell index, yolk colour, eggshell strength, thickness, reflectivity of the eggshell, and surface area.

The eggs (also albumen and yolk) length, width, and height, respectively, were measured by means of an electronic sliding caliper (JOBI profi, XTline, Velké Meziříčí, Czech Republic) with 0.01 mm precision. Total egg weights, yolks, and eggshell weights were determined individually with an electronic balance Ohaus Corporation (Model: Traveler TA502, Parsippany, NJ, USA) with 0.01 g precision. Albumen was carefully absorbed from yolks and eggshells before weighing. Albumen weight was retrieved by subtraction (albumen weight = total egg weight – yolk weight – eggshell weight). The eggshell was dried at room temperature and weighed. After removing the inner membrane, the thickness was measured with an electronic micrometre (precision 0.01 mm; Mitutoyo Corporation, Kawasaki, Japan). Moreover, the diameter and the height of the yolk and albumen was measured. The colour of the yolk was evaluated using the DSM Yolk Colour Fan. The eggshell breaking strength was measured using an Instron Universal Testing Machine (model 3342; Instron Ltd., Norwood, MA, USA) supported by a series Bluehill^®^ Software. Eggshell reflectivity was assessed using an objective photometric method, specifically by a device QCR reflectometer (TSS Chessingham Park, Dunnington, York, England). The measurement is based on the determination of the percentage of light that the surface (eggshell) reflects. The interval of the reflectivity may vary from 0 to 100%. Lower values correspond to darker shades and vice versa. The percentage share of egg components (eggshell, albumen, and yolk) was calculated by using individual egg weight and the weight of the specific egg component. The Haugh unit score, albumen, yolk index, and surface area, according to Kraus et al. [5], and shell index, according to Krunt et al. [1], were also computed

Eggs for the fatty acid determination of egg yolk were collected at 7-day intervals, at the beginning of the first laying cycle at 28, 29, 30 weeks of age (of the hens) and at the beginning of the second laying cycle at 78, 79, and 80 weeks of age. There were always 10 egg yolks collected during the time (a total of 60 yolks for analysis). The yolk samples were pooled and kept frozen at –20 °C until analysis. Analyses of fatty acid profiles were carried out in a laboratory at the Department of Animal Science (Czech University of Life Science Prague, Czech Republic). Methyl esters of fatty acids were analysed following extraction of total lipids according to Folch et al. [19]. Methanolysis was done by applying the catalytic effect of potassium hydroxide and extraction of acids in the form of methyl esters in heptane. The composition of isolated methyl esters was determined by gas chromatograph Master GC (Dani Instruments S.p.A., Cologno Monzese, Italy), with a flame ionization detector and a column with polyethylene glycol as the stationary phase FameWax; 30 m × 0.32 mm × 0.25 μm (Restek Co., Bellefonte, PA, USA), cat. 12,498. A running time of 29 min was used for each sample solution to prevent methyl esters peaks [20]. The following instrumental conditions were employed for GC-FID analysis: injection volume of 1 uL and 5.0 mL/min flow rate of helium carrier gas at a split ratio of 9:1. The injector and detector temperatures used were 200 °C and 220 °C, respectively. The initial oven temperature was maintained at 50 °C for 2 min and then gradually increased to 220 °C at a rate of 10 °C/min, after which it was maintained at 220 °C for 10 min. Clarity software (Version 5.2) was used for the data evaluation and quantification, based on known retention times from a standard Food Industry FAME Mix (Restek Co., Bellefonte, PA, USA).

### 2.3. Calculations

Egg shape index (%) was defined as the ratio between egg length and width multiplied by 100 [21].

Haugh units (HU) were calculated from records of egg weight (in grams) and albumen height (in millimetres) by employing the formula [22] as an indicator of interior egg quality: HU = 100 × log (albumen height, mm − 1.7 egg weight, g^0.37^ + 7.56).

Albumen index (AI, %) was calculated in accordance with Heiman and Carver [23]: AI = (albumen height, mm/(long diameter of albumen, mm + short diameter of albumen, mm)/2) × 100.

Yolk index (YI, %) was computed according to the equation [24]: YI = (yolk height, mm/yolk diameter, mm) × 100.

Yolk to albumen ratio (YAR) was calculated by Dottavio et al. [25]: YAR = (yolk weight, g/albumen weight, g).

Surface area (SA, cm^2^) of each egg was evaluated using the equation reported by Thomson et al. [26]: SA = 4.67 × (egg weight, g)^2/3^.

Shell index (SI, g/100 cm^2^) was calculated using the equation proposed by Ahmed et al. [27]: SI = (shell weight, g/shell surface, cm^2^) × 100.

Egg volume (EV, cm^3^) was determined using the formula EV = π × egg length, mm × (egg width, mm)^2^/6 [28].

From the data on the fatty acid composition, the atherogenic, thrombogenic and saturation index were calculated, which were used for assessing egg lipid quality.

The atherogenic index (AI), indicating the relationship between the sum of the main saturated fatty acids and that of the main classes of the unsaturated, the former considered proatherogenic and the latter antiatherogenic. The atherogenic index was calculated according to Ulbricht and Southgate [29] and Di Lorenzo et al. [30], as follows:AI = [C12:0 + (4 × C14:0) + C16:0]/[∑MUFA + ∑n-6 PUFA + ∑n-3 PUFA],(1)
where C12:0, C14:0, C16:0, MUFA, n-6 PUFA, and n-3 PUFA are the content (percent total FA) of C12:0, C14:0, C16:0, MUFA, n-6 and n-3, respectively.

The thrombogenic index (TI), which shows the tendency to form clots in the blood vessels, can be described as the relationship between the antithrombogenic and the prothrombogenetic (saturated) fatty acids (MUFA, n-6 PUFA, and n-3 PUFA).

The thrombogenic index was calculated in accordance with Ulbricht and Southgate [29] using the formula:TI = [C14:0 + C16:0 + C18:0]/[(0.5 × ∑MUFA) + (0.5 × ∑n-6 PUFA) + (3 × ∑n-3 PUFA) + (∑n-3 PUFA/∑ n-6 PUFA)],(2)
where C14:0, C16:0, C18:0, MUFA, n-6 PUFA, and n-3 PUFA are the content (percent total FA) of C14:0, C16:0, C18:0, MUFA, n-6 PUFA, and n-3 PUFA, respectively.

The saturation index (SI) was calculated using the equation proposed by Ulbricht and Southgate [29]:SI = (C14:0 + C16:0 + C18:0)/(MUFA + PUFA).(3)

The peroxidability index (PI) was calculated according to the equation proposed by Arakawa and Sagai [31]:PI = (% monoenoic × 0.025) + (% dienoic × 1) + (% trienoic × 2) + (% tetraenoic × 4) + (% pentaenoic × 6) + (% hexaenoic × 8).(4)

The amount of each fatty acid was used to calculate the hypocholesterolaemic/hypercholesterolaemic ratio (HHR), as suggested by Santos-Silva et al. [32]:HHR = [(∑C18:1n-9 + C18:2n-6 + C20:4n-6 + C18:3n-3 + C20:5n-3 + C22:5n-3 + C22:6n-3)/(∑C14:0 + C16:0)].(5)

The hypocholesterolemic index (HI) [33] was calculated according to the equation:HI = [(∑C18:1 + C18:2 + C18:3 + C20:3 + C20:4 + C20:5 + C22:4 + C22:6)/(∑C14:0 + C16:0)].(6)

### 2.4. Statistical Analysis

The statistical analysis was processed by the computer application SAS 9.4 (SAS Institute, Inc., Cary, NC, USA). One-way analysis of variance (ANOVA) with the age of hens (1st or 2nd laying cycle) as the fixed factor was used.

Laying cycle effect on each parameter was assessed by the following general linear model:Y_ij_ = µ + L_i_ + e_ij_,(7)
where Y_ij_ was the value of trait, µ was the overall mean, L_i_ was the effect of the laying cycle (first and second), and e_ij_ was the random residual error.

The significance of differences between groups was tested by the t-tests (LSD). The value of *p* ≤ 0.05 was considered significant for all measurements. All data are expressed as mean ± standard deviations. Means marked with a different superscript letter within each column are significantly different. A Pearson correlation analysis (PROC CORR by SAS 9.4) was used to test for correlations.

## 3. Results

In the present study, the differences between some egg quality characteristics and fatty acid compositions of organic eggs, in relation to the ages of laying hens, were investigated.

### 3.1. Egg Quality Characteristics

The results pertaining to some egg quality characteristics are given in Table 2. There were significant differences between laying cycles in egg weight (by 9.89 g), egg surface area (by 7.61 cm^2^), egg volume (by 9.07 cm^3^), yolk proportion (by 1.76 percentage points), yolk to albumen ratio (by 0.04), and shell reflectivity (by 2.01 percentage points) in favour of the second laying cycle. In the first laying cycle, significantly higher results were found in egg shape index (by 2.53 percentage points), albumen proportion (by 1.46 percentage points), albumen index (by 2.25 percentage points), Haugh units (by 8.65), yolk index (by 3.61 percentage points), yolk colour (by 0.55), shell proportion (0.29 percentage points), and shell strength (by 4.17 N.cm^−2^).

### 3.2. Fatty Acid Composition and Indices

The results of the content of saturated fatty acids of organic egg yolks, with regard to laying cycles, are displayed in Table 3. Monounsaturated fatty acids profile of organic eggs yolks is shown in Table 4, polyunsaturated fatty acids are presented in Table 5 and total percentage fatty acid profile of organic eggs yolks and indices related to human health from laying hens at the beginning of the first and second laying cycles are in Table 6. Considering saturated fatty acids, C16:0, C18:0 and C24:0 significantly decreased with the age of the hens and were lower in the second laying cycle by 0.03, 1.25, 0.71 and 0.84 percentage points. On the other hand, monounsaturated fatty acids (C17:1 and C18:1) were significantly higher in the first laying cycle by 0.08 and 6.26 percentage points. Moreover, polyunsaturated acids were also higher in the first laying cycle than in the second laying cycle (C 18:2 by 2.67 percentage points, C 18:3 by 0.04 percentage points, C18:3(9) by 0.31 percentage points, C20:2 by 0.11 percentage points, C20:3 by 0.9 percentage points, C20:4 by 0.22 percentage points). According to the results of the total percentage of the fatty acids profile of organic yolks and indices related to human health, there were significant differences in SFA, which were higher in the first laying cycle than in the second laying cycle by 2.99 percentage points. Furthermore, in the first laying cycle, compared to the second one, there was higher PUFA (by 3.48 percentage points), n-6 PUFA (by 2.92 percentage points), n-3 PUFA (by 0.34 percentage points), atherogenic index (by 0.04), thrombogenic index (by 0.08), saturation index (by 0.06) and peroxidability index (by 4.61). Vice-versa, there was higher MUFA (by 6.43 percentage points), MUFA/SFA (by 0.26), hypocholesterolaemic index (by 0.23), and the hypocholesterolemic/hypercholesterolaemic ratio (by 0.22) in the second laying cycle than in the first laying cycle. Furthermore, correlations were calculated. The significant correlations between the observed parameters were found; they are shown in detail in Table 7. The correlations did not differ between the laying cycles except for the correlation between TI and PI, which was found significant in the second laying cycle and had a negative relationship. Meanwhile the correlation between these two parameters was non-significant in the first laying cycle.

## 4. Discussion

In the present study, the differences between some egg quality characteristics, fatty acid compositions, and human health indices of organic eggs, in relation to the age of laying hens, were investigated.

### 4.1. Egg Quality Characteristics

The positive effect of the flock laying cycle extension was mentioned in a study [6]; the authors concluded that this could be the way to reduce costs, but the age of the hens before depopulating should be carefully considered due to the quality parameters of the eggs. In the present study, two laying cycles were studied; the first one ended by moulting. Regarding to results of egg quality characteristics, the egg weight was significantly higher in the second cycle by 9.89 g, and breaking strength was lower by 4.17 N/m^2^. Indeed, egg volume increased with the weight off eggs [34]. Dunn [35] stated that the handling of larger eggs from old hens is problematic, and they could be more easily broken than lighter eggs due to their lower breaking strength. A higher surface area was also observed in eggs from the older hens. This is not surprising due to the higher egg weight. Considering the yolk, Kowalska et al. [36] described increasing of the yolk proportion with the age, which is in accordance with results of the present study, as well as the higher values of shell reflectivity from older hens.

Moreover, linear increasing of yolk weight and its index with the age of hens was observed by Zita et al. [4]; however, in our study, the reverse trend was reported. Furthermore, yolk colour was higher in the first laying cycle than in the second laying cycle. These results are in accordance with the findings by Rizzi and Chiericato [37], who observed that older hens had a lighter colour of yolks than younger hens. Additionally, the quality of albumen was reduced in the second cycle (e.g., Haugh units decreased from 87.61 to 78.96). Molnár et al. [6] noted that ensuring the supply of eggs of the same quality is crucial for suppliers, but at the end of the cycle could be problematic. That is why the depopulation of flocks should be timed, in a way, when the quality of eggs does not cause economic problems.

### 4.2. Fatty Acid Composition and Indices

In terms of saturated fatty acids, the age (laying cycle, respectively) of hens influenced C16:0, C18:0, and C24:0 fatty acids in yolks. Their amount was lower in the second laying cycle. Considering the C16:0 and C18:0 fatty acids, C16:0 is known for being atherogenic and C18:0 for being neutral to atherogenicity, but instead of that, it is thrombogenic [38]. On the contrary, the research by Qureshi et al. [39] reported that there is no connection between egg intake and blood cholesterol levels. It should be stated that the differences between the fatty acid content in eggs depends on several factors, such as hen genotype [40], diet [41], or feed additives [42]. In terms of the hen’s age, it seems to be important to carefully consider this factor from the point of view of egg quality during the flock management (taking into account the flock’s age during the depopulation). In contrast with saturated fatty acids, monounsaturated fatty acids (C17:1 and C18:1) were higher in the second laying cycle than in the first laying cycle, which is interesting, because other studies (e.g., Lešić et al. [43]) reported that the highest impact of feeding management on C18:1 fatty acid. Moreover, in the study by Koppenol et al. [44], it was found that MUFAs were lower in eggs from younger hens (28- vs. 43- and 58-week-old hens), which is in contrast with our results. They explained their observations with indication of less de novo synthesis or more efficient conversion from a fatty acid precursor to the desired fatty acid in younger hens. Moreover, the differences in the fatty acid composition of egg yolks among the studies may be caused by the use of different hen genotypes, because it was previously confirmed by Rey et al. [2] that genotype significantly affects the fatty acid profiles of egg yolks.

The three major n-3 PUFA (omega-3 polyunsaturated fatty acids), which are connected with health benefits to humans, are α-linolenic acid (ALA), EPA, and DHA. The major role of ALA, as the essential fatty acid, is to provide a substrate for synthesis of long chain n-3 PUFA and DHA [45]. In egg yolks, there are more MUFAs than PUFAs [44]. In the present study, PUFAs were higher in the first laying cycle than in the second laying cycle, as well as the SFAs, which is in accordance with research by Koppenol et al. [44]. On the other hand, Lešić et al. [43] found a higher content of PUFAs in eggs from older hens, but lower SFAs. From a healthy lifestyle point of view, it seems that the eggs from the second laying cycle are of better quality due to their lower atherogenic and thrombogenic indices [46]; however, the hypocholesterolemic index and hypocholesterolaemic/hypercholesterolaemic ratio were higher. The peroxidability index was then higher in eggs from the first laying cycle, which reflects the higher PUFA content, because it was found by Dal Bosco et al. [47] that the higher value resulted in easier oxidation.

Regarding the correlations, in the present study, the same correlations were found between the presented parameters in the first laying cycle and the second laying cycle, except for the correlation between PI and TI. This correlation was found to be significant just in the second laying cycle. However, in the first laying cycle, it was found that this correlation tended to be significant (*p* = 0.096), which could be attributed to the different content of the presence of PUFAs [47] or MUFAs. A negative relationship among SFA, PUFA, HI, and HHR was also found, as well as a positive relationship among SFA, AI, TI, and SI. MUFAs negatively correlated with PUFAs, n-6, and PI. Moreover, PUFAs negatively correlated with AI, TI, SI, and oppositely with n-6, HI, HHR, and PI. Furthermore, n-6 had a negative relationship with AI, TI, and SI, and a positive relationship with HI, HHR, and PI. Meanwhile n-3 just positively correlated with PI. It was also found that AI had a positive relationship with TI and SI, and negatively correlated with HI and HHR. In addition, TI positively correlated with SI and negatively with HI and HHR, while SI had a negative relationship with HI and HHR. In the end, HI positively correlated with HHR. The power of the correlations was also the same in the first and in the second laying cycle. According to indices, AI, TI, and HI were found as factors, which define a healthy animal product. Specifically, low HI reflects a low cholesterol content. With respect to thrombogenicity, stearic acid is considered. Vice-versa, myristic and palmitic acids are considered as atherogenic [48,49]. According to the literature, there is a strong recommendation for humans to follow diets with atherogenic, thrombogenic, and hypercholesterolemic foods, in terms of eliminating atherosclerosis and the risk of cardiovascular disorders [50].

## 5. Conclusions

The present study shows that there is an impact—of the hen’s age—on the internal and external egg quality in organic rearing systems. Younger hens had a higher albumen, yolk index, Haugh units, and eggshell strength, while the MUFAs were lower and SFAs were higher. The saturation, atherogenic, and thrombogenic indices were significantly higher in the egg yolks from older hens compared to younger layers. According to these results, it seems clear that the usage of hens, for two laying cycles, is appropriate (in terms of egg quality), particularly if human health and diet is contextually discussed. This study was the first to compare two laying cycles of organically reared hens, in terms of fatty acid profile changes. Other research should be conducted to balance the suitable ages of hens at the end of the cycles.

## Figures and Tables

**Table 1 animals-12-00125-t001:** Chemical composition of diet and its major fatty acids profile.

Characteristic	Nutrient Composition (%)
Crude protein	17.00
Crude fat	5.00
Crude fibre	6.00
Ash	13.10
Methionine	0.32
Lysine	0.80
Calcium	3.80
Phosphorus	0.55
Sodium	0.16
ME (MJ/kg)	11.0
	FA ^1^ (% of total FA)
C14:0 ^2^	0.19
C16:0 ^3^	13.84
C18:0 ^4^	3.98
C20:0 ^5^	0.35
C22:0 ^6^	0.18
C16:1 n-7 ^7^	0.21
C18:1 n-9 ^8^	27.41
C20:1 n-9 ^9^	0.61
C18:2 n-6 ^10^	51.64
C18:3 n-3 ^11^	4.30
C18:3 n-6 ^12^	0.38
C20:4 ^13^	0.21
SFA ^14^	18.07
MUFA ^15^	27.41
n-6 PUFA ^16^	52.02
n-3 PUFA ^17^	4.30
n-6/n-3 PUFA	12.32

^1^ FA, fatty acids; ^2^ C14:0, myristic FA; ^3^ C16:0, palmitic FA; ^4^ C18:0, stearic FA; ^5^ C20:0, arachidic FA; ^6^ C22:0, behenic FA; ^7^ C16:1 n-7, palmitoleic FA; ^8^ C18:1 n-9, oleic FA; ^9^ C20:1 n-9, gadoleic FA; ^10^ C18:2 n-6, linoleic FA; ^11^ C18:3 n-3, α-linolenic; ^12^ C18:3 n-6, γ-linolenic FA; ^13^ C20:4, arachidonic FA; ^14^ SFA, saturated fatty acid; ^15^ MUFA, monounsaturated fatty acid; ^16^ n-6 PUFA, omega 6 polyunsaturated fatty acid; ^17^ n-3 PUFA, omega 3 polyunsaturated fatty acid.

**Table 2 animals-12-00125-t002:** Some quality characteristics of organic eggs from (of) laying hens at the beginning of the first and second laying cycles (mean ± SD ^1^).

Parameter	Laying Cycle		SEM ^2^	*p*-Value
	1st	2nd		
Egg weight (g)	61.21 ^b^ ± 4.39	71.10 ^a^ ± 5.64	0.527	0.0001
Egg shape index (%)	78.69 ^a^ ± 2.20	76.16 ^b^ ± 3.38	0.232	0.0001
Egg surface area (cm^2^)	72.49 ^b^ ± 3.47	80.10 ^a^ ± 4.26	0.405	0.0001
Egg volume (cm^3^)	56.48 ^b^ ± 3.90	65.55 ^a^ ± 6.13	0.510	0.0001
Albumen proportion (%)	66.18 ^a^ ± 2.18	64.72 ^b^ ± 2.78	0.194	0.0001
Albumen index (%)	10.29 ^a^ ± 2.38	8.04 ^b^ ± 1.80	0.178	0.0001
Haugh unit	87.61 ^a^ ± 8.34	78.96 ^b^ ± 8.55	0.706	0.0001
Yolk proportion (%)	23.70 ^b^ ± 1.99	25.46 ^a^ ± 2.45	0.178	0.0001
Yolk index (%)	47.63 ^a^ ± 3.31	44.02 ^b^ ± 2.63	0.260	0.0001
Yolk to albumen ratio	0.36 ^b^ ± 0.05	0.40 ^a^ ± 0.05	0.004	0.0001
Yolk colour (DSM scale fan)	8.77 ^a^ ± 1.48	8.22 ^b^ ± 0.95	0.094	0.0037
Shell proportion (%)	10.12 ^a^ ± 0.80	9.83 ^b^ ± 0.90	0.064	0.0210
Shell thickness (mm)	0.338 ± 0.03	0.336 ± 0.03	0.002	0.6500
Shell strength (N.cm^−2^)	46.65 ^a^ ± 7.28	42.48 ^b^ ± 9.75	0.658	0.0014
Shell reflectivity (%)	28.48 ^b^ ± 4.52	30.49 ^a^ ± 5.01	0.363	0.0052
Shell index (g.100 cm^−2^)	8.53 ± 0.64	8.70 ± 0.71	0.051	0.0968

^1^ SD, standard deviation; ^2^ SEM, standard error of the mean; ^ab^data bearing different letters in the same row are significantly different (*p* ≤ 0.05).

**Table 3 animals-12-00125-t003:** Saturated fatty acids profile (% of total FA ^1^) of organic eggs yolks from laying hens at the beginning of the first and second laying cycles (mean ± SD ^2^).

Parameter	Laying cycle		SEM ^3^	*p*-Value
	1st	2nd		
Myristic (C14:0)	0.35 ± 0.07	0.32 ± 0.05	0.0097	0.1894
Palmitic (C16:0)	26.95 ^a^ ± 1.34	25.70 ^b^ ± 1.00	0.2102	0.0019
Margaric (C17:0)	0.26 ± 0.08	0.24 ± 0.05	0.0103	0.4128
Stearic (C18:0)	9.64 ^a^ ± 1.08	8.93 ^b^ ± 0.61	0.1480	0.0144
Behenic (C22:0)	0.11 ± 0.25	0.02 ± 0.04	0.0291	0.1007
Lignoceric (C24:0)	1.02 ^a^ ± 1.00	0.18 ^b^ ± 0.48	0.1399	0.0017

^1^ FA, fatty acid; ^2^ SD, standard deviation; ^3^ SEM, standard error of the mean; ^ab^ data bearing different letters in the same row are significantly different (*p* ≤ 0.05).

**Table 4 animals-12-00125-t004:** Monounsaturated fatty acids profile (% of total FA ^1^) of organic eggs yolks from laying hens at the beginning of the first and second laying cycles (mean ± SD ^2^).

Parameter	Laying Cycle		SEM ^3^	*p*-Value
	1st	2nd		
Palmitoleic (C16:1)	2.24 ± 0.62	2.38 ± 0.48	0.0871	0.4244
Margaroleic (C17:1)	0.09 ^b^ ± 0.06	0.17 ^a^ ± 0.05	0.0106	0.0003
Oleic (C18:1)	30.48 ^b^ ± 1.58	36.74 ^a^ ± 1.67	0.5626	0.0001
Gadoleic (C20:1)	0.30 ± 0.06	0.29 ± 0.06	0.0095	0.5436

^1^ FA, fatty acid; ^2^ SD, standard deviation; ^3^ SEM, standard error of the mean; ^ab^ data bearing different letters in the same row are significantly different (*p* ≤ 0.05).

**Table 5 animals-12-00125-t005:** Polyunsaturated fatty acids profile (% of total FA ^1^) of organic egg yolks from laying hens at the beginning of the first and second laying cycles (mean ± SD ^2^).

Parameter	Laying Cycle		SEM ^3^	*p*-Value
	1st	2nd		
Linoleic (C18:2)	23.71 ^a^ ± 2.62	21.04 ^b^ ± 2.20	0.4341	0.0013
γ-Linolenic (ω-6) (C18:3)	0.13 ^a^ ± 0.02	0.09 ^b^ ± 0.04	0.0061	0.0002
α-Linolenic (ω-3) (C18:3(9))	1.04 ^a^ ± 0.11	0.73 ^b^ ± 0.13	0.0310	0.0001
Eicosadienoic (C20:2)	0.33 ^a^ ± 0.07	0.22 ^b^ ± 0.04	0.0121	0.0001
Eicosatrienoic (C20:3)	0.22 ^a^ ± 0.05	0.13 ^b^ ± 0.03	0.0104	0.0001
Arachidonic (C20:4)	2.11 ^a^ ± 0.31	1.89 ^b^ ± 0.25	0.0470	0.0201

^1^ FA, fatty acid; ^2^ SD, standard deviation; ^3^ SEM, standard error of the mean; ^ab^data bearing different letters in the same row are significantly different (*p* ≤ 0.05).

**Table 6 animals-12-00125-t006:** Total percentage of fatty acid profiles of organic eggs yolks and indices related to human health from laying hens at the beginning of the first and second laying cycles (mean ± SD ^1^).

Parameter	Laying cycle		SEM ^2^	*p*-Value
	1st	2nd		
SFA ^3^ (% of total FA ^4^)	38.53 ^a^ ± 1.51	35.54 ^b^ ± 1.39	0.3300	0.0001
MUFA ^5^ (% of total FA)	33.26 ^b^ ± 1.90	39.69 ^a^ ± 1.76	0.5890	0.0001
PUFA ^6^ (% of total FA)	28.16 ^a^ ± 2.48	24.68 ^b^ ± 2.15	0.4570	0.0001
n-6 PUFA ^7^ (% of total FA)	25.95 ^a^ ± 2.45	23.03 ^b^ ± 2.14	0.4290	0.0003
n-3 PUFA ^8^ (% of total FA)	1.64 ^a^ ± 0.36	1.30 ^b^ ± 0.26	0.0560	0.0015
MUFA/SFA	0.86 ^b^ ± 0.06	1.12 ^a^ ± 0.07	0.0226	0.0001
PUFA/SFA	0.73 ± 0.09	0.70 ± 0.08	0.0132	0.1645
n-6/n-3 PUFA	16.56 ± 3.99	18.24 ± 3.35	0.5910	0.1587
AI ^9^	0.46 ^a^ ± 0.04	0.42 ^b^ ± 0.03	0.0061	0.0002
TI ^10^	1.07 ^a^ ± 0.08	0.99 ^b^ ± 0.05	0.0124	0.0007
SI ^11^	0.60 ^a^ ± 0.05	0.54 ^b^ ± 0.03	0.0080	0.0001
HI ^12^	2.08 ^b^ ± 0.17	2.31 ^a^ ± 0.15	0.0309	0.0001
HHR ^13^	2.10 ^b^ ± 0.17	2.32 ^a^ ± 0.15	0.0310	0.0001
PI ^14^	40.93 ^a^ ± 3.54	36.32 ^b^ ± 2.67	0.6131	0.0001

^1^ SD, standard deviation; ^2^ SEM, standard error of the mean; ^3^ SFA, saturated fatty acids; ^4^ FA, fatty acid; ^5^ MUFA, monounsaturated fatty acid; ^6^ PUFA, polyunsaturated fatty acid; ^7^ n-6, omega 6 PUFA; ^8^ n-3, omega 3 PUFA; ^9^ AI, atherogenic index; ^10^ TI, thrombogenic index; ^11^ SI, saturation index; ^12^ HI, hypocholesterolemic index; ^13^ HHR, hypocholesterolaemic/hypercholesterolaemic ratio; ^14^ PI, peroxidability index; ^ab^ data bearing different letters in the same row are significantly different (*p* ≤ 0.05).

**Table 7 animals-12-00125-t007:** Correlation among lipids profiles, atherogenic, thrombogenic, and hypocholesterolemic indices at the beginning of the first (above the diagonal) and second (below the diagonal) laying cycles.

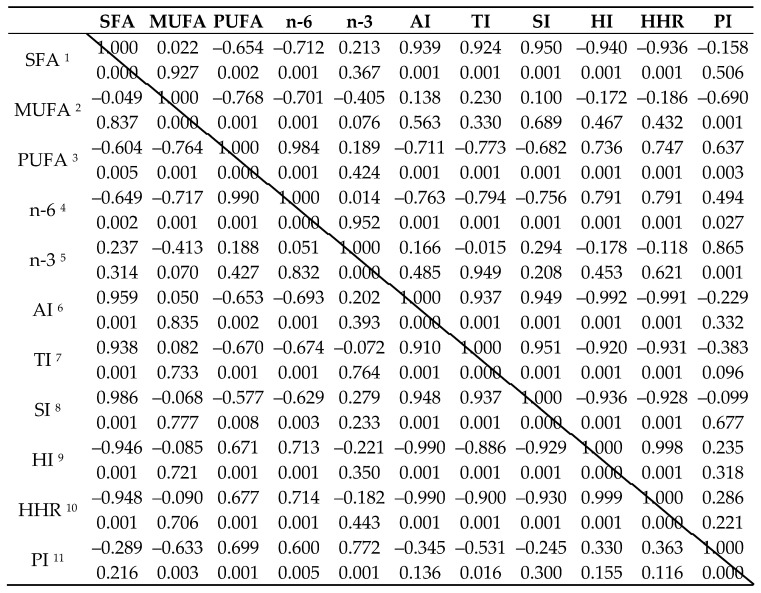

^1^ SFA, saturated fatty acid; ^2^ MUFA, monounsaturated fatty acid; ^3^ PUFA, polyunsaturated fatty acid; ^4^ n-6, omega 6 FA; ^5^ n-3, n-3 PUFA; ^6^ AI, atherogenic index; ^7^ TI, thrombogenic index; ^8^ SI, saturation index; ^9^ HI, hypocholesterolemic index; ^10^ HHR, hypocholesterolaemic/hypercholesterolaemic ratio; ^11^ PI, peroxidability index.

## Data Availability

The data presented in this study are available upon reasonable request from the corresponding author.

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
