# Peer review of "Changes in Fatty Acids Profile, Health Indices, and Physical Characteristics of Organic Eggs from Laying Hens at the Beginning of the First and Second Laying Cycles"

_animals, 2022, doi:10.3390/ani12010125_

Round 1
Reviewer 1 Report
Changes in fatty acids profile, health indexes and physical 2 characteristics of organic eggs from laying hens at the begin- 3 ning of the first and second laying cycle 4 Lukáš Zita 1, *, Monika Okrouhlá 1 , OndÅ™ej Krunt 1 , Adam Kraus 1 , LudÄ›k Stádník 1 , Jaroslav ÄŒítek 1 and Roman 5 Stupka 1
General comments
The paper studies the effect of the age of hens on some quality parameters of eggs. The study provides novel and interesting information, but it need of further revisions. The introduction section needs to improve writing and style in order to be clearer for readers. Material and methods also needs to be improved. My main concern is that authors miss important information, such as nutrient requirement according to hens age. Hence, authors don not indicate if feeding fulfill the nutrient requirements of birds, which could affect on egg qualty. Also some information according to methods description needs to be included. In addition some mistakes have been identify in results section and discussion needs also of improvement.
Introduction
Lines 38-40. “Nowadays, there is a trend to follow different or alternative ways [1], how to rear 39 laying hens with respect to their welfare and the quality of eggs, which are an integral 40 part of the human diet.”
Write the first sentence again
Line 42 “in many ways, such as by genotype [2],” This work is more on the housing system. Use a reference specify for the genetic effect. Ex Rey et al. https://doi.org/10.3390/ani11071944
Lines 45-46 “Considering 44 the housing of hens, there were 55.6 % of hens in enriched cages, 25.7 % housed on deep 45 litter, 14.1 % in free range systems and 4.6 % were reared in organic systems [7].”
This data are for Europe? or for which country refers to?
Lines 52-53 “The content of fatty acids is about 26.6 g/100 g of yolk”. Please include reference
Lines 58-59 The egg is naturally poor in 58 linolenic acid and does not contain eicosapentaenoic (EPA) and docosahexaenoic (DHA) 59 fatty acids.” But you said before that poultry eggs are one of the best sources of n-3.
Please clarify. You You should include here that eggs can be enriched by n-3
Lines 65-66. You must include information on the posibilities for n-3 enrichement for example organic systems
Ex. Lopez-Bote et al., Effect of free-range feeding on ny3 fatty acid and
a-tocopherol content and oxidative stability of eggs. Animal Feed Science Technology 72 1998 33–40
Lines 67-68. “However, to our knowledge, there are not too many reports in the available literature 67 regarding the effects of age of laying hens reared in organic farming systems on the fatty 68 acid profile and indexes related to human health in its egg” You dont say anything before on the organic system.
You should conect this paragraph with the n-3 fatty acid enrichement possibilities in the organic system
Material and methods.
Lines 79-90. You dont say if the feeding fulfill the nutrient requirements of birds. Was the diet similar for whole breeding period. Requirements would probably change by age and if bird were not properly fed this could affect on the egg qualty. Please clarify.
Lines 112-113. Please include the trade mark of the balance and supplier.
Lines 118-119-. How was the diameter and the height of youlk and albumen measured?. Please indicate
Lines 126-127. “The Haugh units score, albumen and yolk index, surface area and shell 127 index were also computed.”. Indicate how were they measured.
Lines 138-143. “The contents of isolated methyl esters were determined using a gas chromato- 138 graph Master GC (Dani Instruments S.p.A., Cologno Monzese, Italy) equipped with a 139 flame ionization detector and a column with polyethylene glycol as the stationary phase 140 (FameWax; 30 m × 0.32 mm × 0.25 μm). Helium was used as the carrier gas, with a flow 141 rate of 5 ml/min and a split ratio of 1:9. The obtained records were analyzed using Clarity 142 software, Version 5.2 and quantified on the basis of known retention times from a stand- 143 ard Food Industry FAME Mix (Restek Co., Bellefonte, USA).”
Include the chromatographic conditions for samples injection and the column used.
Line 185. Why did you use this hypocholesterolemi index (reference is for ham??).
Results
Lines 217-218 “The results of the content of saturated fatty acids of organic egg yolks with regards 217 to laying cycles are displayed in Table 2”. Change table 2 by table 3
Lines 219-221. “Monounsaturated fatty acids profile of organic 218 eggs yolks is shown in Table 3, polyunsaturated fatty acids are presented in Table 4 and 219 total percentage fatty acid profile of organic eggs yolks and indexes related to human 220 health from laying hens at the beginning of the first and second laying cycle are in Table 221 5”
Please check the table number, there are some mistake.
Lines 222. “C15:0, C16:0, C18:0 and C24:0 significantly decreased”. But C15:0 is not presented in table.
Tables. Please present first SEM and then Pvalue in the tables.
Discussion
Lines 288- 296. Authors need to discuss further the effect of yolk color, quality of albumen…etc
Line 299. Check content, in tables C15:0 is not presented
Lines 304-312. Were the birds fed following the nutrient requeriments?. If not, fatty acid composition of yolk could be affected. Please clarify.
Lines 312-315. Could be the hens line affect on the results on fatty acid. See paper Rey et al. https://doi.org/10.3390/ani11071944. Where birds of same genetic type. Explain in the text.
Conclusions
Lines 354-356. “The n-6/n-3 PUFA ratio, saturation, atherogenic and thrombogenic indices 355 were significantly lower in the egg yolks from older hens compared to younger layers”. Please check n-6/n-3 PUFA ratio was not statistically affected…and the others?. Check with information that you provide in tables.
Author Response
Dear reviewers,
First of all, on behalf of all authors, I would like to thank to both reviewers for valuable comments and recommendations. All the comments were carefully considered, and the manuscript was corrected according to these comments to improve the final quality of the manuscript. The correction of the manuscript was made to satisfy all reviewers as much as possible. We believe that the corrections that were made are sufficient and will satisfy your requirements. Detailed answers to the general comments and specific comments are listed below in this document.
Sincerely yours, Lukáš Zita

Reviewer 2 Report
-Simple summary: indicate the meaning of the acronyms MUFA and SFA, and PUFAn-6 / n-3
-Results: 3.2. Table 2 is Table 3, Table 3 is Table 4; Table 4 is Table 5; Table 5 is Table 6.
In table 3, Where does your C15 Que appear in the text ????
S / P indicate text in table 6, or in text indicate YES
Table 7 is difficult to interpret with the naked eye, could another format be applied?
Discussion: -I believe that what is indicated between lines 330-348 should be set out in the results section and not in the discussion section, so this should be related to results obtained by other authors.
Author Response

(The authors gave the same response as above.)

Round 2
Reviewer 1 Report
Changes proposed have been properly made, and questions have been answered, so I consider the paper has now improved.